# The Complete Mitochondrial Genome of One Breeding Strain of Asian Swamp Eel (*Monopterus albus*, Zuiew 1793) Using PacBio and Illumina Sequencing Technologies and Phylogenetic Analysis in Synbranchiformes

**DOI:** 10.3390/genes12101567

**Published:** 2021-10-01

**Authors:** Haifeng Tian, Qiaomu Hu, Hongyi Lu, Zhong Li

**Affiliations:** 1Yangtze River Fisheries Research Institute, Chinese Academy of Fishery Sciences, Wuhan 430223, China; tianhf@yfi.ac.cn (H.T.); qmhu@yfi.ac.cn (Q.H.); luhy99@163.com (H.L.); 2College of Fisheries, Huazhong Agricultural University, Wuhan 430070, China

**Keywords:** *Monopterus albus*, swamp eel, mitochondrial genome, PacBio, phylogenetic analysis, Synbranchiformes

## Abstract

Asian swamp eel (*Monopterus albus*, Zuiew 1793) is a commercially important fish due to its nutritional value in Eastern and Southeastern Asia. One local strain of *M. albus* distributed in the Jianghan Plain of China has been subjected to a selection breeding program because of its preferred body color and superiority of growth and fecundity. Some members of the genus *Monopterus* have been reclassified into other genera recently. These classifications require further phylogenetic analyses. In this study, the complete mitochondrial genomes of the breeds of *M. albus* were decoded using both PacBio and Illumina sequencing technologies, then phylogenetic analyses were carried out, including sampling of *M. albus* at five different sites and 14 species of Synbranchiformes with complete mitochondrial genomes. The total length of the mitogenome is 16,621 bp, which is one nucleotide shorter than that of four mitogenomes of *M. albus* sampled from four provinces in China, as well as one with an unknown sampling site. The gene content, gene order, and overall base compositions are almost identical to the five reported ones. The results of maximum likelihood (ML) and Bayesian inference analyses of the complete mitochondrial genome and 13 protein-coding genes (PCGs) were consistent. The phylogenetic trees indicated that the selecting breed formed the deepest branch in the clade of all Asian swamp eels, confirmed the phylogenetic relationships of four genera of the family Synbranchidae, also providing systematic phylogenetic relationships for the order Synbranchiformes. The divergence time analyses showed that all Asian swamp eels diverged about 0.49 million years ago (MYA) and their common ancestor split from other species about 45.96 MYA in the middle of the Miocene epoch. Altogether, the complete mitogenome of this breed of *M. albus* would serve as an important dataset for germplasm identification and breeding programs for this species, in addition to providing great help in identifying the phylogenetic relationships of the order Synbranchiformes.

## 1. Introduction

Swamp eels (family Synbranchidae) are widely distributed in western Africa, Liberia, Asia, the Indo-Australian Archipelago, Mexico, and Central and South America. Most of them are freshwater fishes of the tropics and subtropics, which are classified into two subfamilies and four genera [1]. One Asian swamp eel, *Monopterus albus* (Zuiew, 1793), which can be found in China, Japan, and Southeast Asia, has been popularly cultured in central and southern China [2], Vietnam [3], and countries of Southeast Asia [4] due to its flavor and nutritional profile, while another local strain, which is mainly farmed in the Jianghan Plain, Hubei Province, China, has been subjected to consecutive selective breeding programs recently due to its preferred body color and superior growth rate and fecundity [5,6]. Interestingly, at least five to six different phylogenetic lineages have been found in China based on mitochondrial markers recently [7,8], and at least five cryptic species were found throughout East and Southeast Asia when more samples from Indonesia were included [9,10]. Additionally, *M. cuchia* (Hamilton, 1822), one species that is widely distributed in South Asia and farmed in Bangladesh [11], has been reclassified as *Ophichthys* alongside with five other species (*M. desilvai*, *M. hodgarti*, *M. fossorius*, *M. ichthyophoides*, and *M. indicus*), while some other members have been resurrected to the genera *Rakthamichthys* (*M. digressus*, *M. eapeni*, *M. rongsaw*, and *M. roseni*) and *Typhlosynbranchus* (*M. boueti* and *M. luticolus*) based on osteology [12]. Moreover, the monophyly of the order Synbranchiformes seems to be controversial, as Indostomidae (*Indostomus paradoxus*) was classified as Gasterosteiformes and has been previously revealed to group with the Synbranchidae family [1,13,14]. Obviously, such taxonomical modifications of the Synbranchidae family reflect the great progress made in this species group, which lacks significant external morphological characters, while the phylogenetic relationship between the genus *Monopterus*, family Synbranchidae, and order Synbranchiformes still needs to be confirmed, as only limited markers and limited species were included in these analyses [12,15,16].

Mitochondrial genomes (mitogenomes) of animals are generally 16–19 kb long circular molecules. Due to their maternal inheritance, short coalescence times, unambiguous orthology, and rapid evolutionary rates [17], mitogenomes have been widely used in phylogeny [18], taxonomic resolution [19], and animal phylogeography [20]. 

Recently, we have sequenced the whole genome of the same breed of *M. albus* [21]. Although the mitogenomes of *M. albus* sampled from four provinces of China [22] and one with unknown sampling site [18] have been reported, we have decoded the complete mitogenome of this breed using PacBio S Single Molecule Real-Time sequencing (SMRT) long reads and high-precision Illumina sequencing technologies, and hope this data would contribute to germplasm identification in the future. Besides, the phylogenetic relationship of the selected breed, plus five *M. albus* isolates from other origins, and species of order Synbranchiformes with complete mitogenome available were investigated by using the Maximum likelihood (ML) and Bayesian analyses, and their divergence time were also estimated. These results can serve as important resources for further studies and will provide more insights into the phylogeny and evolution of Synbranchiformes.

## 2. Materials and Methods

### 2.1. Sampling, Sequencing, Genome Assembly, and Annotation 

The *M. albus* sample was collected from a fish farm in Xiantao (113.38 E, 30.23 N), located at the Jianghan Plain in Hubei Province, China. The experimental procedure was approved by the Animal Experimental Ethical Inspection of Laboratory Animal Centre, Yangtze River Fisheries Research Institute, Chinese Academy of Fishery Sciences (ID Number: 2020-THF-02). The muscle sample was immediately frozen in liquid nitrogen after collection, and then stored at −80 °C in the Breeding Laboratory, Yangtze River Fisheries Research Institute (Sample cod is HBXT-20191006). DNA was isolated using the conventional phenol chloroform method. The preparation of one 20 kb SMRTbell library for PacBio sequencing and one pair-end library for Illumina sequencing and sequencing method were described in previous report [21].

The PacBio subreads for the mitochondria genome were filtered out with pbmm2 (https://github.com/PacificBiosciences/pbmm2, on 16 June 2021) using one previously published reference mitochondrial genome (accession no.: KP779623); then, filtered reads were assembled de novo using Canu v1.8 [23]. The mitochondrial assembly was first improved using the alignment of all long reads with pbmm2 and polished using gcpp2 (with default parameters; https://github.com/PacificBiosciences/gcpp, accessed on 16 June 2021). Subsequently, the circularity of the assembly was checked with Circlator v1.5.5 [24]. Then, a final round of polishing was performed with gcpp2 (with default parameters), as mentioned before. In addition, the mitochondrial genomes were assembled from Illumina short-read data with NOVOPlasty 4.2.1 [25] using a published COI sequence (accession: KP779623) for the query. The assembled mitogenome was annotated using MitoAnnotator on the MitoFish webserver (http://mitofish.aori.u-tokyo.ac.jp/annotation/input.html) (version 3.67; accessed 18 July 2021) [26].

### 2.2. Phylogenetic Analysis, Topology Testing, and Molecular Dating

To further investigate the phylogenetic position of our breed and the phylogenetic relationship of Synbranchiformes, a total of 19 complete mitogenomes of the order Synbranchiformes were obtained through searching against the nonredundant (nr) database from the National Center for Biotechnology Information (NCBI) database using blastn and by searching the nucleotide database using the keywords “Synbranchiformes” and “mitochondrion”. The species *Channa gachua* (GenBank accession no. NC_036948) of the order Anabantiformes was used as an outgroup. Two datasets were constructed, whereby the first one was concatenated from 13 protein-coding genes (PCGs), 22 tRNAs, 2 rRNAs, and a D-loop region and included 16249 sites; the second one only consisted of 13 PCGs and included 11,556 sites. These were parsed, aligned, trimmed, and concatenated through PhyloSuite [27]. The best-fitting substitution models and partitioning schemes were selected in PartitionFinder v2.1.1 using the Bayesian information criterion [28]. The Bayesian inference (BI) tree was conducted with MrBayes 3.2.6 [29]. Two independent Markov chain Monte Carlo (MCMC) chains were run for 2 × 10^6^ generations, and the convergence between two runs was assessed when the standard deviation of the split frequencies was below 0.01. The first 25% of sampled data was discarded as “burn-in”, then the rests were used to generate a majority rule consensus tree. The maximum likelihood (ML) tree was reconstructed using IQTREE 1.5.5 software [30]. The best fitting model was selected using ModelFinder [31] implemented in IQ-TREE, since it has more sequence evolution models than PartitionFinder. The node support was evaluated using 1000 ultrafast bootstrap replicates (UFBS) [32]. To investigate the relationship of our breed with the phylogenetic lineages reported previously, the D-loop sequences of our breed and the 167 D-loop sequences sampled from different locations throughout China [7] were analyzed with DNASP [33] and the haplotypes were subjected to a constructed neighbor-joining (NJ) tree using MEGA version 7 [34], as described previously [7].

To test the phylogenetic position of *Indostomus paradoxusa*, hypothetical trees grafting *I. paradoxusa* as the sister group to Synbranchiforme and Mastacembelidae clades were generated using Mesquite v3.40 [35]. To confirm the taxonomic problem of spiny eel as revealed by Chu et al. [36], the constraint monophyly of spiny eel (*Sinobdella sinensis* (MZ188892.1 and KP342509.1)) and lesser spiny eel (*Macrognathus aculeatus* (KF636363.1 and KT443991.1)) species was estimated in IQ-TREE using the ‘-g’ constraint option and concatenated whole datasets. Then, tests were performed in IQ-TREE for site log likelihoods using the concatenated datasets and “-au” option by specifying 1000 bootstrap replicates.

Considering the presence of inter- and intraspecies in the phylogenetic tree [37,38], divergence times were estimated using the RelTime method implemented in Mega X [39] with the GTR + G + I modeling. Two calibration time points for Mastacembelus (the common ancestor of *Mastacembelus erythrotaenia* and *M. armatus*, 18.56 to 32.44 MYA) and Synbranchiformes (common ancestor to *I. paradoxus* and *M. erythrotaenia*, 66.8 to 79.4 MYA) from the TimeTree web resource (http://www.timetree.org/, accessed on 25 August 2021) were used [40]. The Bayesian phylogenetic tree was used to estimate the divergence times to ensure consistency between the divergence times and phylogenetic tree. The resulting time tree was plotted using the packages ‘ggtree’ [41] and ‘deeptime’ (https://github.com/willgearty/deeptime, accessed on 25 August 2021).

## 3. Results

### 3.1. Mitochondrial Genome Assembly and Annotation

After filtering, 440 subreads were obtained and two sequences of 30,323 bp and 52,972 bp were assembled using canu. The first one being circulated was kept and submitted for the following analysis. After circulation and two rounds of polishing, a circular DNA sample of 16,621 bp was obtained. Additionally, one totally identical assembly was obtained using Illumina paired reads, indicating the accuracy of the obtained assembly. This newly determined mtDNA sequence from the present study was deposited in GenBank with the accession number MZ597543 (Figure 1).

The length of the obtained mitogenome was 16,621 bp, which was only one nucleotide shorter than those five full mitogenomes reported previously [18,22]. The overall sequence similarity was in the range of 97–99%. The overall nucleotide composition was 28.92% A, 27.11% T, 14.50% C, and 29.47% G, with a slight A and T bias of 56.03%. This mitogenome consisted of 13 PCGs, 2 rRNAs, 22 tRNAs, and one putative control region (D-loop). Most of the genes were encoded on the H-strand, with only ND6 and eight tRNAs (Gln, Ala, Asn, Cys, Tyr, Ser, Glu, and Pro) located on the L-strand. The structural organization of this mitogenome conformed to the five previously reported ones for *M. albus* and the other teleost [22,42].

Ten protein-coding genes (ND1, ND2, COII, ATP8, ATP6, COIII, ND4L, ND5, ND6, and CYTB) had the ATG start codon, two protein-coding genes (COI, ND4) had the GTG start codon, and ND3 had ATA as the start codon. Three protein-coding genes (ND1, ATP8, ND5) had complete TAG stop codons, COI had AGG as the stop codon, and two protein-coding genes (ND4L and ND6) had TAA stop codons. Two protein-coding genes (ND2, ATP6) had TA(A) as the stop codon and five protein-coding genes (COII, COIII, ND3, ND4 and CYTB) had T(AA) as the stop codon. The AT- and GC-skews for the concatenated PCGs and whole mitogenomes are summarized in Appendix A. Interestingly, a similar phenomenon was found, whereby AT-skews did not conform to the whole genome in all six swamp eels, as observed in other teleosts [43].

The amino acid (AA) codon usages were assessed by calculating relative synonymous codon usage (RSCU) values in 13 PCGs (Figure 2A). A total of 3807 codons were encoded by 13 PCGs, and the most frequently used codons were CUA (5.1%), CUC (4.4%), and AUU (4.3%). The analysis of the AA components and their codon usage revealed that the codons encoding Leu^(CUN)^ represented over 100 codons per thousand (CDpT), followed by Thr and Ala, while those encoding Cys were rare (Figure 2B). The patterns of AA codon usage and RSCU were consistent with the other five mitochondrial genomes reported previously (Appendix A).

### 3.2. Phylogenetic Analyses

The topologies of the ML trees and Bayesian trees obtained using two datasets were almost identical, except for some inconsistency of the internal relationships of all six sequences of *M**. albus* (Appendix A). The phylogenetic tree obtained using 13 PCGs is shown in Figure 3. In the phylogenetic tree, all Asian swamp eels were clustered together as a monophyletic clade. The selected breed derived from the Jianghan Plain branched firstly in this clade, then this clade grouped with a clade including the genera *Ophisternon* and *Synbranchus*, while the genus *Ophichthys* was recovered at the earliest branch in the clade consisting of family Synbranchidae. When the D-loop of the selected breed and the sequences reported by Cai et al. were subjected to haplotype analyses, all six *M. albus* breeds were recovered in lineage C (Appendix A). Additionally, all species of the family Mastacembelidae were recovered in a separate monophyletic clade and the genus *Mastacembelus* was grouped as a monophyletic subclade, although two lesser spiny eels (*M. aculeatus*) and two spiny eels (*S. sinensis*) were mixed. Furthermore, *I. paradoxus* was recovered as the outgroup of Synbranchidae and Mastacembelidae. The hypothetical trees constraining *Indostomus* as the sister group to Synbranchidae (*p* value for AU test: 0.0118) or Mastacembelidae (*p* value for AU test: 0.0035) were significantly rejected (Table 1). The constraint monophyly of two spiny eels and two lesser spiny eel was also rejected (*p* value for AU test: 0.0047).

The time tree (Figure 4) was inferred with RelTime due to the presence of mixed sampling in our dataset and the good performance. Based on our results, the selected breed and other wild individuals began diverging about 0.49 million years ago (MYA), while the genus *Monopterus* split from the common ancestor of genera *Synbranchus* and *Ophisternon* about 45.96 MYA, within the Eocene epoch. The estimated split age for *Ophisternon* and *Synbranchus* was about 36.9 MYA, while the split age for Synbranchidae and Mastacembelidae was traced back to about 64.1 MYA, within the Paleocene epoch.

## 4. Discussion

In this work, we have decoded the complete mitogenome of one selected breed of *M. albus* derived from the Jianghan Plain using PacBio long reads and Illumina short reads. This breed and the other five isolates were classified to the largest lineage C [7], which is consistent with populations derived from the Yangtze River. The newly obtained mitogenome is only one nucleotide shorter than those reported previously, and the gene content, gene order, overall base compositions, and sequence identity are almost identical to the five previously reported ones. Interestingly, this breed was revealed to be the deepest branch in the clade consisting of swamp eels from China, suggesting that our breed derived from the Jianghan Plain is different from those sampled from other provinces in China. Considering the many different phylogenetic lineages and great amount of trade of seedings of *M. albus* between different provinces in China, further work is required to develop breed- or population-specific markers for germplasm purity identification and artificial breeding. The data obtained here would provide fundamental data for future germplasm identification, collection, and breeding programs for this species.

The classification of the genus *Monopterus* has been greatly modified based on the osteology and COI [12,44]. In this study, based on the whole mitogenome sequence, the phylogenetic relationships of four genera of Synbranchidae were confirmed, which were generally consistent with the results based on the morphological characteristics and COI [12]. As complete mitogenomes of species of the genus *Monopterus* (e.g., *M. bicolor*, *M. dienbienensis*) and other genera (e.g., *Typhlosynbranchus*, *Rakthamichthys*, and *Macrotrema* ) are still lacking, the detailed phylogenetic relationships of the family Synbranchidae remain to be validated in future. Notably, *I. paradoxus* was not grouped with Synbranchidae, as reported recently [13,14], and corresponding hypothetical trees constrained as sister groups to Synbranchidae or Mastacembelidae were both rejected significantly by the AU test (Table 1). Considering that the phylogenetic position of Indostomoidei is still under debate [1,14], these results indicate that Indostomoidei diverged earlier than the other two members of the order Synbranchiformes. Additionally, two spiny eels isolated from the Qian Tang River (MZ188892) and Yangtze River (KP342509) were not clustered as a monophyletic group and the hypothetical constraint tree was also rejected by the AU test (Table 1). This is consistent with the phylogenetic analysis reported by Chu et al. [36], suggesting that the classification of the spiny eel should be studied further. The divergence times estimated based on phylogenetic trees using the RelTime method showed that the estimated time for all Asian swamp eels was about 0.49 MYA, while the genus *Monopterus* was differentiated from other members by about 45.96 MYA within the middle Eocene epoch. The estimated split between *Ophisternon* and *Synbranchus* occurred at 36.9 Mya within the Eocene epoch, which is slightly earlier than previous estimations based on CYTB b and ATPase 8/6 genes [45]. The divergence time of the Synbranchidae and Mastacembelidae families could be dated back to 64.1 Mya within the Paleocene epoch. With further mitogenome data for this order, these results would provide a consolidated framework for further studies, such as the karyotype or chromosome evolution [46,47,48], adaptive evolution to different environments [49] for those closely related species, and the dispersal history of the order Synbranchiformes [7,45] in the future.

## Figures and Tables

**Figure 1 genes-12-01567-f001:**
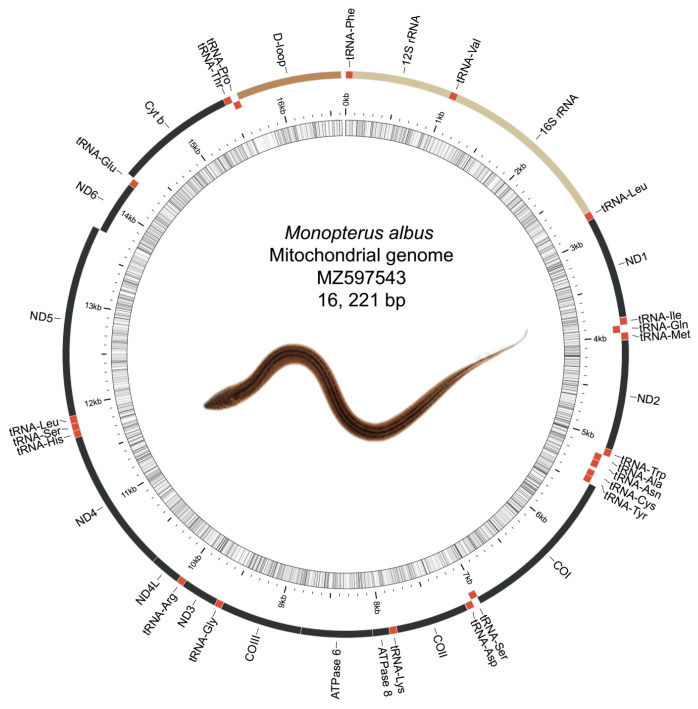
Mitochondrial genome map of *M. albus* breeding strain generated with MitoAnnotator. PCGs, rRNAs, tRNAs, and the D-loop are shown in different colors. Protein-coding genes are shown in black; tRNAs shown in red and designated by their three letter abbreviations. Light brown represents the rRNAs, while dark brown represents the D-loop. Genes located at the H- or L-strand are mapped outside or inside of the circle. The innermost circle of the images represents (G + C)% for every 5 bp of the mitogenome; the darker the line, the higher the (G + C)%.

**Figure 2 genes-12-01567-f002:**
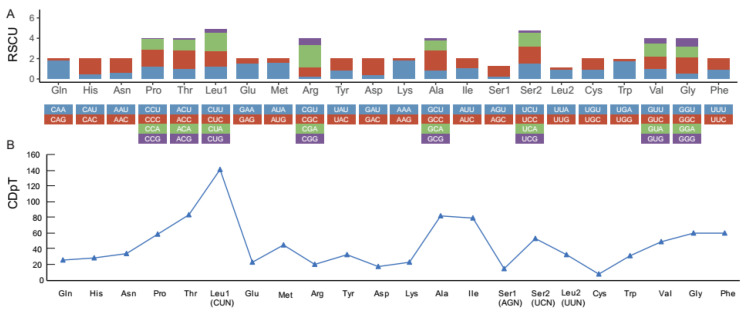
The relative synonymous codon usage (RSCU) (**A**) and codon distribution (**B**) of PCGs in the mitogenomes of selected breeds of *M. albus*. CDpT, codons per thousand codons.

**Figure 3 genes-12-01567-f003:**
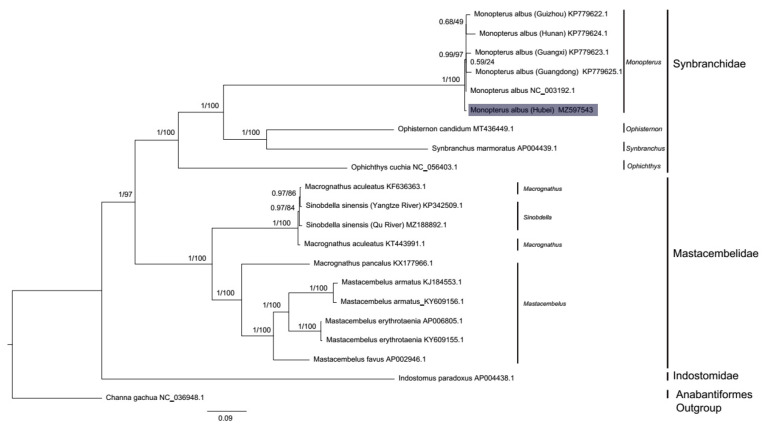
The phylogenetic tree for the order Synbranchiformes inferred from the 13 concatenated PCGs of 20 complete mitochondrial sequences obtained using MrBayes v.3.2.6 and IQ-TREE v.1.5.5. *Channa gachua* (NC_036948.1) was used as the outgroup. The new obtained mitogenome of *M. albus* breed is shown with grey background. The numbers at branches are Bayesian posterior probabilities and ultrafast bootstrap support values, respectively. GenBank accession numbers are given after the scientific name.

**Figure 4 genes-12-01567-f004:**
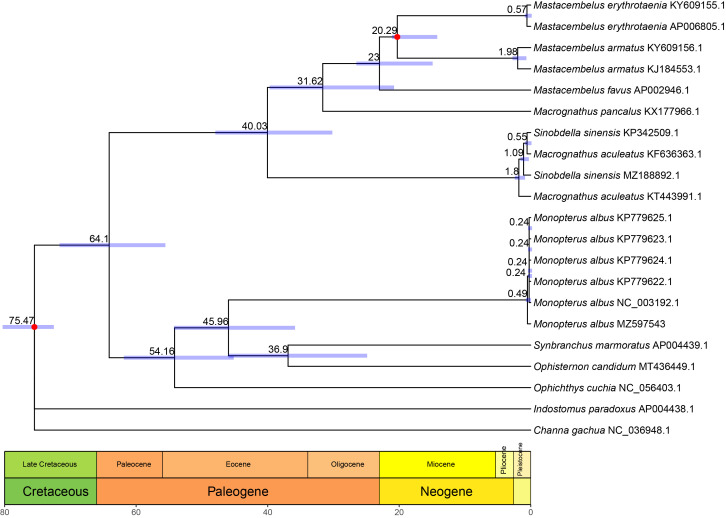
The evolutionary timeline of the order Synbranchiformes was estimated using RelTime based on the 13 concatenated PCGs. Numbers on the scale bar are millions of years before the present. Light blue bars on nodes represent 95% confidence intervals. The red dot denotes the calibration time point. Geologic timescale is shown at the bottom of the tree.

**Table 1 genes-12-01567-t001:** Constraint topology test results from IQ-TREE.

	deltaL	bp-RELL	p-KH	p-SH	p-WKH	p-WSH	c-ELW	p-AU
Unconstrained	0	0.988	0.98	1	0.98	1	0.9875	0.9927
*Indostomus paradoxus* grouped with Synbranchidae	31.339	−0.011	−0.02	+0.093	−0.02	−0.0330	−0.0112	−0.0118
*Indostomus paradoxus* grouped with Mastacembelidae	37.324	−0.001	−0.003	−0.047	−0.003	−0.004	−0.0013	−0.0035
Monophyly of *Sinobdella sinensis* and *Macrognathus aculeatus*	71.940	0	0	−0.002	0	0	0	−0.0047

## Data Availability

The data that support the findings of this study are openly available in NCBI Genbank under the accession no. MZ597543. The raw sequence data reported in this paper have been deposited in the Genome Sequence Archive in the National Genomics Data Center, Beijing Institute of Genomics (China National Center for Bioinformation), Chinese Academy of Sciences, under accession no. CRA003062, which are publicly accessible at https://bigd.big.ac.cn/gsa, accessed on 20 December 2020.

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
