# Peer review of "The Complete Mitochondrial Genome of One Breeding Strain of Asian Swamp Eel (Monopterus albus, Zuiew 1793) Using PacBio and Illumina Sequencing Technologies and Phylogenetic Analysis in Synbranchiformes"

_genes, 2021, doi:10.3390/genes12101567_

Round 1

Reviewer 1 Report

The MS 1400092 by Tian et al reports original data on the mitogenome of a breeding strain of the Asian swamp eel Monopterus albus. These data were used for a phylogenetic reconstruction of the order Synbranchiformes and to estimate the divergence time of the families/genera/species. In my opinion, two main pitfalls can be identified in the structure of the paper and they should be fixed. The first one concerns the scope of the paper, which should be better defined at the end of the introduction. The second one is that although previous literature is cited and complete mitogenome of other species are included, there is no attempt to link present data to previous: it is not clear if is there any relationships between the breed analysed in this study and previous 5-6 different lineages identified in China on the base of Control region (CAI, YU, MIPAM, ZHANG and YUE 2013, included in the reference list). The point is that breeding strains should be related to natural populations (otherwise which is the meaning of a phylogenetic reconstruction?). Finally, I am not a native English speaking but I could easily identify some language mistakes (as an example at lines 159 and 222). I suggest a language check or revision by a mother language.

See here below my point by point suggestions

lines 29-30. Here it is not clear in which way these data will help in germplasm conservation and breeding programs. This is cited also at the end of the introduction but is never explained.

lines 66-68 better define which is the scope of the study

Lines 112-113: why you decided to test the monophyly of spiny eel Sinobdella sinensis and lesser spiny eel Macrognathus aculeatus? There was a specific taxonomic problem with these species (Chu et al. 2020) and you should more deeply describe it in the introduction and/or in the scope of the study. In addition, the unresolved relationship was confirmed after having built your tree, so to better analyse your results you correctly decided to apply the test for monophyly. Then you should better explain it in the materials and methods

line 157: delete “interestingly”

line 159 “was not conforms” change to “did not conform”

line 176 “the selecting breed derived from Jianghan plain ….” You never mention before this locality, and this creates confusion in readers that are not from China and are not familiar with the geography of the area. You may add this information before, at line 71.

lines 176-178 Can you indicate which of the CR Chinese haplotype have the six Monopterus albus included in your analyses? Using as a reference those identified  by Cai et al 2013

line 222: “are still remain to be validated” change to “remain to be validated”

Reference list: I suggest a careful revision because there are many mistakes on capital letters and italics. At line 261 and later along with the reference list the name of the species should be changed to Monopterus albus

Line 275: Is this a book? Information on publisher and number of pages are missing

Reviewer 2 Report

The authors successfully sequenced the mitogenome of Monopterus albus and used to study the phylogeny of several members of the family Synbranchidae. The methodology is adequate, and I agree with the authors when they state that this work can be a useful resource for future research. However, the authors do not really discuss the implications of their findings. For example, why is the breed sequenced so different from the remaining members of the same species? What does it mean for the use of genetic resources of this fish? Also, the authors could further explore how their results can be used to study the evolution of the order Synbranchiformes.

Several grammatical errors were found, and I suggest the authors to revise the language of the manuscript. Bellow I point out some of these errors and other small inconsistencies.

Introduction:

A paragraph or just a few sentences explaining the importance of having the sequence of the mitogenome as a resource will help this manuscript.

Line 36: “refer to family Synbranchidae”

Line 40: “that is widely distributed”

Line 49: Replace “to genus” by “as” and “along with other five ones” by “alongside five other species”.

Line 51: “to the genus”

Line 53: “the order”

Line 54: “that was classified as” instead of “that classified in order” and change “previously has been revealed” to “it has been previously revealed”

Line 58: remove “are” from “are still need”

Line 62: “of the same breed”

Line 63: Change this sentence to something like: The phylogenetic relationships of the selected breed, plus five M. albus isolates from other places, and species of the order Synbranchiformes with complete mitochondrial genome available were investigated by using the Maximum likelihood (ML) and Bayesian analyses.”

Line 66: Change “could” by “can”

Line 67: Change “an important data set of the germplasm resources” to “an important resource” and “study” to “studies”

Line 68: “the phylogeny”

Material and methods

Line 78: Reference to pbmm2 is missing

Line 81: Reference to gcpp is missing. Was the polishing made with long reads? Also, the authors mention polishing was done on the mitochondrial assemblies of both species. Which species are these? Isn’t this paragraph referring to the assembly of the mitogenome of M. alubs only? Please clarify.

Line 85: Change “using reported COI” to “using a published COI sequence”

Line 86: Which annotation method? Do the authors mean that annotation was conducted using MitoAnnotator?

Line 97: “consisted” and replace “that including” by “and included”.

Line 103: change to “was below…”

Line 104: change “were” to “was”

Line 106: Delete “When analyzed using IQ-TREE”

Line 107: Replace “as it has more” by “since it has more”

Line 108: replace “by“ by “using”

Line 115: The analysis concatenated whole datasets? Pleas clarify.

Results

Line 129: Change “and the first one suggested to be circular” to “being the first one circular”

Line 130: “of polishing”

Line 174: Which difference?

Line 182: “the outgroup”

Discussion

Line 208: “selected breed” instead of “selecting breed”

Line 209: add “and” after “PacBio long reads”

Line 217-218: variation in the COI marker

Line 221: “of the genus”

Line 225: “constrained“  instead of “constraining”
